# Evaluating profitability of beef cattle farming and its determinants among smallholder beef cattle farmers in the Baljovan District of Khatlon region, Tajikistan

**Farrukh Jobirov**[1]*, **Zhang Yuejie**[1], **Cornel Anyisile Kibona**[1,2]

**1** College of Economics and Management, Jilin Agricultural University, Changchun, Jilin, China,
**2** Department of Agricultural Economics and Finance, Mwalimu Julius. K. Nyerere University of Agriculture and Technology, Musoma, Tanzania

* jobirovfarrukh@gmail.com

**Data Availability Statement:** All relevant data are within the paper and its Supporting Information files.

## Abstract

In Tajikistan, owning beef cattle is an important survival mechanism for smallholder farmers to alleviate poverty. Therefore, beef cattle farming enterprises should indeed strive to maximize profit to excel and flourish in a free economy. Nevertheless, smallholder beef cattle farmers are known for making little profit. Thus, this study was set to evaluate the profitability of beef cattle farming and its determinants to enhance profit maximization among smallholder beef cattle farmers in the Baljovan District of Khatlon region, Tajikistan. A total of 388 farming households were chosen at random and purposive for the study. The cross-sectional data collected using questionnaires was analyzed by using descriptive, gross margin (GM), and ordinary least squares (OLS) regression models. Based on the descriptive analyses, the mean age of beef cattle farmers was 52.73 years, with a household size of 7.07 members. The beef cattle farmers had an average of 18.23 cattle herd size with 8.54 years of farming experience. The average land area possessed by farmers was 10.59 hectares. Among farmers, men (98.2%) dominated beef cattle farming activities. Around 83.8% of farmers had a college grade (higher literacy). Besides, around 89.4% of farmers had access to farm credits. However, only 71.4% of farmers used farm credit points to produce beef cattle. Most of the farmers (89.7%) had access to accurate market information. Such market information enabled 75.8% of farmers to sell their beef cattle to open market (profitable) outlets rather than middlemen. About 89.4% had access to veterinary services. Additionally, about 82.7% of farmers acknowledged the availability of pasture for grazing, which motivated 87.6% of farmers to be involved in selling contracts. Furthermore, economic investigation results revealed that on average, farmers had a gross margin (GM-profit) of 353.77 US$ per cattle, with feed costs (58.6%) and medications costs (26.1%) accounting for the largest share of total variable costs. Meanwhile, the profitability of beef cattle farming among farmers was significantly influenced by education level, family size, farming experience, pasture availability, land size owned, selling contract, feed costs, medications expenses, access to credits, and sales costs (P < 0.05). This study concluded that beef cattle production is a feasible business. However, the potential for increased profitability is significant if existing

**Funding:** Yes; This research is supported and funded by the National Beef Cattle Industrial Technology System and Industrial Economy Research Project under the Ministry of Agriculture in the People's Republic of China (PRC) (CARS-37). The funders had no role in study design, data collection and analysis, decision to publish, or preparation of the manuscript.

**Competing interests:** The authors have declared that no competing interests exist.

resources are efficiently coordinated and production expenses, notably feed and healthcare costs, are minimized. Thus, the government should develop additional measures for addressing concerns such as capacity building, suitable and freely available pasture as well as health management, to boost beef cattle profitability among farmers in Tajikistan.

## 1. Introduction

In Tajikistan, owning beef cattle is an important survival mechanism for smallholder farmers [1, 2]. Beef cattle are kept by families for several purposes, which vary based on their prosperity, environment, and endowment. Beef cattle give numerous advantages to millions of agricultural households in underdeveloped countries, from monetary revenue to food, from dung to draft power in farming, benefiting people's lives via multiple channels [1]. Beef cattle, in particular, can contribute to prosperity by generating income and in-kind revenue from the sale of cattle and/or the marketing and utilization of mammal foodstuffs including meat and milk [1]. Furthermore, cattle serve as a safety net in the form of liquid assets, and it is commonly regarded also as a form of investment and security attributed to the fact that the sale of cattle offers an immediate cash influx to cope with unanticipated financial instability [3, 4]. Additionally, owning cattle can facilitate access to legal and tacit loans since they can be used as collateral [5, 6].

Tajikistan's turbulent history has had a negative influence on the agriculture sector's current situation [2]. Its passage from the Soviet Union to independence had been an arduous one, hampered by a five-year civil conflict that damaged infrastructures and interrupted marketplace links. Deforestation and overgrazing have worsened soil quality and raised the danger (threats) of soil degradation, rock slides, and floods [2]. The threats have arisen as a result of low infrastructure improvements, a lack of suitable laws and policy frameworks for grazing and forest management, and restricted institutional support. This uncertainty has a significant negative impact on the beef cattle industry [2]. However, according to a study published by the Republic of Tajikistan [2], beef cattle stocks have increased to levels greater than in the immediate pre-independence period, and beef cattle husbandry is an operation in which practically almost every poor population participates. Because of the increase in stocks accompanied by the decrease in feed availability, feed per beef cattle has declined considerably, as has beef cattle performance. There are numerous barriers to the advancement of the beef cattle industry, including a lack of human resources, underprivileged pasture governance, poor planning of community beef cattle, an absence of feed during the cold season, environmental devastation, and limited access to healthy forages grains and infrastructural facilities, all of which are exacerbated by climate change [2].

Investment initiatives to address the challenges of improving beef cattle efficiencies will have an important influence on smallholders' dietary diversity, food security status, revenue, jobs, and livelihood opportunities [7, 8]. Due to the general intense relationship between beef cattle efficiency and relevant use of pastures, a concomitant focus on grazing strategic planning components of beef cattle production is required [8]. The accessibility of wintertime fodder plus early summer pastures, which do not fulfill requirements, are the production constraining variables. Summertime grasses, on the other hand, supplied substantial grazing excess compared to present requirements. As a result, there has been a periodic imbalance between supply and demand [7, 8]. Steadily increasing beef cattle herds in certain areas have put further strain on improperly organized communal grazing. Because the growth period for calves after weaning frequently relies on this grassland and grazing on other peripheral sites, this adds to feeder

calves' poor health and productivity [9]. Despite increasing beef cattle populations, relatively low ownership implies that beef cattle production is still mostly a subsistence operation [9].

Tajikistan produces the majority of its meat from bulls born to mixed-breed cattle with a focus on dairy qualities. There are various particular beef breeds in the nation; however, the majority of beef cattle are native breeds (local breeds) mixed with dual-purpose breeds like Brown Swiss or Brown Carpathian [10].

The Tajik government recognized the need to re-establish grassland and minimize deterioration to increase pasture farmland for sustainable beef cattle production. To achieve such an initiative, the Tajik government imposed the Livestock and Pasture Development Project I & II (LPDP I & II), which was supported by the International Fund for Agricultural Development (IFAD), from 2015 to 2017 [2]. The initiative addressed around 38,000 farmers in at least 200 settlements throughout five districts of the Khatlon region, all of which have significant poverty rates and the opportunity for improved beef cattle productivity. The project's key target respondents included: (i) smallholder livestock communities; (ii) commercial veterinarian service providers and micro enterprises with the ability to assist rural families and local farmers; and (iii) women-headed families and impoverished women. The project's overall purpose was to alleviate poverty in Khatlon Region. Moreover, the explicit purpose was to improve the nutritional status and income of about 38,000 impoverished families by increasing beef cattle output [2].

According to the Republic of Tajikistan's press release [2], the investments and operations of Livestock and Pasture Development Project I & II (LPDP I & II) were carried out for three main purposes. The first was to realize institutional advancement by strengthening the public sector and civic groups, such as through developing reliable and successful prop-poor grassland monitoring systems. The second was to improve animal health and performance by improving access to veterinary care, which resulted in lower morbidity and improved herd performance. The third purpose was to achieve grassland advancement and expansion for disaster mitigation by improving access to more capable and climate-resilient grassland zones, as well as diverse revenue possibilities for beef cattle societies, through self-sustaining, community-led biodiversity conservation. The major aim was to improve the living standards of beef cattle farmers in selected districts of the Khatlon region and the nation as a whole.

Despite efforts to increase the economic viability of Tajikistan's beef cattle industry and thereby eradicate poverty, Tajikistan remains a food-insecure state, having 46% of Tajiks living in extreme poverty [11]. Poverty is especially severe in rural parts, where the people are mostly reliant on farming, livestock, and remittances for a living and food supply [11]. This demonstrates the poor economic gains, effectiveness, and modernization level of beef cattle production in Tajikistan, notably in the Khatlon region, where the project was carried out. Additionally, despite 27 years of solid economic progress, Tajikistan is still the poorest and least industrialized nation in Central Asia. The economy is still in its early stage, with a low value-added (including productivity improvement to beef cattle) and a limited export foundation [11].

Profit maximization is one of the primary goals of every business company (beef cattle farming enterprise) for the long performance and sustainability [12, 13]. Profitability is the act of profit-earning abilities, which is a significant aspect of a company's survival (beef cattle farming business) [12, 13]. Furthermore, profit has a substantial impact on the company's achievement of other business future such as economic and social development, technology, jobs, and technical advancement [12, 13].

Beef cattle farming enterprises should indeed strive to maximize profit to excel and flourish in a free economy, or they will be forced into bankruptcy for failing to create adequate money [14]. Beef cattle farming enterprises are having difficulty achieving the requisite profit due to

increased competitive pressures and inefficiency. As a result, the issue of what variables influence profitability should be a top concern for academicians and researchers, comprising decision-makers, investment companies, debtors, and policymakers dealing with beef cattle farming businesses [13].

The study of the factors of profitability has increased in popularity throughout time in a variety of research fields. Experts in business strategy, economics, and accounting believe that a company's internal resources have a substantial influence on its profitability [12, 15, 16]. Generally, profitability (performance) is an important measure of the effectiveness of any dividend enterprise (profit-oriented beef cattle farming enterprise). The enterprise (beef cattle farming enterprise) goal (profitability) is achieved as an outcome of the effective utilization of the set of resources (determinants) for the aim of maximizing income to the greatest extent feasible.

There are almost no recorded studies that have examined the profitability of Tajikistan's beef cattle farming enterprise among smallholder farmers. Therefore, this study was set to assess the economic benefits of beef cattle farming and its determinants among beef cattle farmers in the Baljovan District of Khatlon region, Tajikistan, specifically to (i) analyze the socioeconomic characteristics of beef cattle farmers and (ii) examine the profitability of beef cattle farming and its determinants among smallholder beef cattle farmers. Thus, the purpose of this study was to convey an awareness of how beef cattle farming (business) performs and what it contributes to the economy of the smallholder beef cattle farmer in Tajikistan for poverty reduction.

## 2. Materials and methods

### 2.1. Ethical considerations

All ethical issues were addressed appropriately during the information-gathering procedures. The Jilin Agricultural University Graduate Research Ethics Committee in China originally authorized this research. The Ministry of Agriculture then authorized it for the order id (RP.2021/89/25). Thereafter, initially, the respondents' consent was acquired orally, and then the accurate information in the consent letter was conveyed to all targeted respondents. Before their involvement in the case, the respondents were invited to fill out and execute the documents as verification of their agreement to engage in the survey. The consent was then authorized by the Zone Executive Officer (ZEO). All respondents were assured that they had the option to withdraw from the event at any stage.

### 2.2. Description of the study area

The study was conducted in the Baljovan District of the Khatlon Region. The region and its district were selected for this study because Tajikistan implemented Livestock and Pasture Development Project I & II (LPDP I &II). Khatlon region, particularly the selected district has a potential for increased beef cattle productivity, with a high poverty level. The Khatlon region is located in the southwest of the country, with a population of 2.64 million people, or approximately one-third of the country's total population.

### 2.3. Sampling procedures and sample size

Purposive and randomized sampling was used to select participants at different phases, with a sample size of 388 beef cattle farmers generated by a multistage stage sampling technique. In the first stage, the Khatlon Region zone was purposively selected from five major beef cattle-producing regions in Tajikistan. In the second stage, Baljovan District was purposively selected

**Table 1. Distribution of the sample size.**

| District | Villages | Population | Percentage Proportion | Sample |
|---|---|---|---|---|
| Baljovan | Baljyvon | 2,571 | 20.15 | 78 |
| | Tojikiston | 2,540 | 19.90 | 77 |
| | Sayf Rahim | 2,555 | 20.02 | 78 |
| | SafarAmirshoev | 2,560 | 20.06 | 78 |
| | Sari Hosor | 2,535 | 19.87 | 77 |
| **Total** | | **12,761** | **100** | **388** |

from five districts of the Khatlon Region covered by Livestock and Pasture Development Project I & II (LPDP I &II). The selected district has great potential beef cattle-producing areas. In the third stage, five villages (strata) (Baljyvon, Tojikiston, Sayf Rahim, Safar Amirshoev, and Sari Hosor), were randomly selected. Finally, the number of respondents (sample size) selected from each village (stratum) was determined by utilizing the percentage proportion (see Table 1). This study applied Slovin's formula [17] to the targeted (N = 12,761) beef cattle farmers to determine a randomly selected sample size of 388 representative beef cattle farmers. The Slovin's formula [17] as applied by Kibona and Yuejie [18] is mathematically expressed as;

$$n = \frac{N}{1 + Ne^2} = \frac{12,761}{1 + 12,761(0.05)^2} = 387.81 \approx 388 \tag{1}$$

Whereby $N$ is the targeted population size, $n$ is a sample size, and $e$ is the error tolerance level or is the level of precision provided by Yamane [19] to determine the required sample size at a 95% confidence level and 90% level of precision.

## 2.4. Techniques for data collection

The structured questionnaire and interview approaches were used to gather cross-sectional primary data from smallholder beef cattle farmers. The structured questionnaires and interviews collected data on (i) socioeconomic characteristics of smallholder beef cattle farmers; (ii) expenses of beef cattle production (variable costs) and revenue; (iii) beef cattle ownership status (including breed types and breeding methods); (iv) sales of beef cattle (marketing); distinctively, the market outlets (the distribution channels), the number of beef cattle sold, the selling price, and the contract selling; and (v) availability of pasture and coping strategies during severe outbreaks of cattle diseases and shortage of pasture.

## 2.5. Theoretical framework

The theory of Resource Based Views (RBV) provided the framework for this study. According to Barney [12] and Wernerfelt [20], the RBV theory is a concept that considers resources to be critical to a better company (better beef cattle farming enterprise) performance (profitability). The resources allow the business (beef cattle farming business) to achieve and maintain a competitive edge. To be useful, resources ought to be beneficial, unique, and pricey to imitate (costly), as well as better controlled.

Beneficial: As per Wernerfelt [20], resources are beneficial if they assist firms (beef cattle farming businesses) in maximizing the price provided to consumers. This is accomplished via enhancing distinctiveness and/or lowering input costs (minimizing the cost of farming). Resources that are unable to achieve this need are at a competitive disadvantage. Unique: As per Curran et al., [21], and Wernerfelt [20], unique resources are those that can be obtained on an exceptional basis by one or a few enterprises (beef cattle farming enterprises). Competitive

balance occurs when many firms (beef cattle farming enterprises) have the same resources or competence [12, 20]. Costly: If an enterprise (beef cattle farming enterprise) wishes to maintain a competitive edge, the resource ought to be costly to replicate or substitute for a competitor. Therefore, resources alone do not provide a competitive edge to an enterprise (beef cattle farming enterprise) if they are not controlled to maximize their worth. Only the enterprise that can utilize beneficial, unique, and costly imitate resources will be able to maintain a competitive edge in the long run [20, 22].

The Resources Based View (RBV) analyzes company performance in different ways, such as explaining profitability primarily based on distinct firm-level attributes, resources, and competencies [20, 23]. As per the Recourses Based View concept, enterprises take divergent historical pathways, resulting in diverse competency that impacts their capacities in various ways [20, 23]. Progressive companies (beef cattle farming enterprises) in a sector are effective since they have access to a variety of resources and use them efficiently to obtain a competitive edge. Along with this perspective, resources include all internal and external resources such as brand names (beef cattle farming method), strengths, credits, stock (herds size), cash (capital-variable costs), skilled people (education level), location (distance to the market), an appropriate management system, entity good products, information (access to market information), and skills, and even cost-effective strategies that are moderately connected to the enterprise (beef cattle farming enterprise) [20]. As a result, the functional purpose of each enterprise (beef cattle farming enterprise) is to form its distinctive mix of resources to strengthen its economic capability, resulting in increased profit (profit maximization) [12, 24]. The concept is relevant in this research because it demonstrates how an enterprise uses its internal resources, as well as its capacity to interact with the surroundings, to generate greater profit. In addition, internal resources such as wealth, skilled workforce, and expert knowledge, as well as the ability to access the market, credits, and market information through trading with the treasury, socializing with customers on social media, and other abilities such as company size, age of the firm, and firm site, are the primary resources that are linked with the determinants of profitability. In this study, determinants of profitability were investigated using resources-based theory. As a result, there is a strong relationship between theory and determinants of profitability.

## 2.6. Data analysis

Both descriptive statistical analysis and econometric models were used in this study. The information gathered through questionnaires and interviews was coded and analyzed using Excel and SPSS v. 22.

**2.6.1. Descriptive data analysis.** Descriptive statistical analysis involving means, maximum, minimum, standard deviations, frequencies, and percentages were used to examine the: (i) socioeconomic characteristics; (ii) beef cattle ownership status; (iii) market outlets (the distribution channels); (iv) selling price; (v) contract selling; and (vi) availability of pasture.

**2.6.2. Economic data analysis models.** The economic data were analyzed using a gross margin analysis (GMA) and ordinary least squares (OLS) multiple linear regression models as described below;

*2.6.2.1. The gross margin analysis model.* The gross margin analysis model was used to examine the profitability of beef cattle farming among smallholder beef cattle farmers. The gross margin (GM) was computed by subtracting the total revenue (TR) from the total variable cost of beef cattle farming (TVC). This formula is formally described as follows:

$$GM_i = TR_i - TVC_i \tag{2}$$

Where, $TR_i$ stands for the total revenue of farming per beef cattle in US \$, $TVC_i$ is the total variable cost per beef cattle in US \$, and $GM_i$ represents the gross margin of farming per beef cattle in US\$. In addition, according to Kibona and Yuejie, [25], to discover profit value, it is necessary to first determine physical values and unit prices for input and output variables, followed by a computation of producing expenses and revenues. As a result, the functional equation for the GMA model is described below.

$$\text{GM} = \sum_{i=1}^{n} Y_i Py_i - \sum_{j=1}^{m} X_j Px_j \tag{3}$$

Here; $GM$ stands for the gross margin per beef cattle, $\sum_{i=1}^{n} Y_i Py_i$ is the total revenue (TR) of $n$ beef cattle, $Py_i$ is the beef cattle market price, $Y_i$ is the total number of beef cattle sold, $\sum_{j=1}^{m} X_j Px_j$ is the total cost of $m$ variable inputs per beef cattle, $X_j$ is the amount of the $j^{th}$ variable input (j = 1, 2, 3. . .$n$, $m$ inputs), $Px_j$ is the cost of inputs utilized per unit/price per unit of a variable input, and $\Sigma$ is the summation symbol.

Additionally, the TVCs were calculated by adding the expenses of husbandry labor (excluding family labor costs), medicines (treatment), feeds (silage, fodder, and other feeds like soybean meal and urea), mineral salts and vitamins, spraying or dipping, veterinary services, and some other expenses for sales and logistics. Furthermore, for this study, fixed costs were not included in gross margin estimates since fixed costs are non-quantifiable owing to the traditional managerial style utilized by smallholder beef cattle farmers [25].

*2.6.2.2. The Ordinary Least Squares multiple linear regression model.* The Ordinary Least Squares (OLS) model was used to analyze the determinants of profitability of the beef cattle farming business among smallholder beef cattle farmers. The model is precisely specified as follows:

$$Y_i = \beta_0 + \beta_1 X_1 + \beta_2 X_2 + \beta_3 X_3 + \ldots\ldots\ldots\ldots + \beta_n X_n + \varepsilon_i \tag{4}$$

Where
Yi = gross margin-GM (profit) of i$^{th}$ smallholder beef cattle farmer per beef cattle (US\$)
$\beta_0$ = constant
$\beta_1$,. . .,.. $\beta_n$ = coefficients to be estimated
$X_1$,. . .., $X_n$ = independent variables
$\varepsilon_i$ = error term, this represents all factors that influence the variance but are not depicted by the explanatory variables [26].

As cited by Kibona and Yuejie, [18], the OLS approach is a numerical modeling tool that is used to explain the relationship between a continuous dependent variable (gross margin-profit) and several independent factors (determinants of profitability) [27]. The multiple linear regression approach was used to determine the intensity and significance of the relationship between profitability per beef cattle and factors that are anticipated to affect profitability [26]. These include the variable costs (feed costs, husbandry labor costs, medicines (treatment) costs, and sales and logistics costs), education level of a farmer, family size, cattle herd size, land owned, experience in farming, access to veterinary services, access to credits, pasture availability, selling contract, marketing channels, and access to market information.

**Table 2. The pre-hypothesized sign effects of the independent variables on beef cattle profitability.**

| Variables | Measurement | Hypothesized sign effects |
|---|---|---|
| Feed costs | US$ | - |
| Husbandry labor costs | US$ | - |
| Medicines(treatment) costs | US$ | - |
| Sales and logistics costs | US$ | - |
| Education level | Years of schooling | + |
| Family size | Number of members | + |
| Land owned | In hectare | + |
| Cattle herd size | Number of beef cattle | + |
| Experience in farming | In years | + |
| Access to veterinary services | If 0 = No, 1 = Yes | + |
| Access to credits | If 0 = No, 1 = Yes | + |
| Pasture availability | If 0 = No, 1 = Yes | + |
| Selling contract | If 0 = No, 1 = Yes | ± |
| Marketing channels (market outlets) | If 1 = Open market, 2 = butcheries; 3 = Tajikistan Meat Commission; 4 = Middlemen | ± |
| Access to market information | If 0 = No, 1 = Yes | + |

Therefore, the OLS regression model for the determinants of profitability of the beef cattle farming industry among smallholder beef cattle farmers was further described as follows:

$$
\begin{aligned}
Gross\ margin\ (profit) &= \beta_0 + \beta_1 Feed\ costs + \beta_2 Husbandry\ labour\ costs + \beta_3\ Medicines\ costs \\
&+ \beta_4\ Sales\ and\ logistic\ costs + \beta_5 Education\ level + \beta_6 Familiy\ size + \beta_7 Cattle\ herd\ size \\
&+ \beta_8 Land\ owned + \beta_9 Experience\ in\ farming + \beta_{10} Access\ to\ veterinary\ services \\
&+ \beta_{11} Access\ to\ credits + \beta_{12} Pasture\ availability + \beta_{13} Selling\ contract \\
&+ \beta_{14} Marketing\ channels + \beta_{15} Acces\ to\ market\ information + \varepsilon(5)
\end{aligned}
$$

According to Mlote et al., [26], a failure of the assumptions underlying OLS regression analysis may threaten the validity of the regression results. Any failure of the hypotheses (explanatory variables not having a normal distribution, autocorrelation, heteroscedasticity, and multicollinearity) renders the component estimations invalid for interpretation. Therefore, the model was verified for validity using SPSS v. 22 statistical analyses. The Variance Inflation Factor (VIF) resulted in a score of 1.31, confirming the absence of multicollinearity between the dependent and independent variables [28]. Table 2 below presents the pre-hypothesized sign effects of the independent variables on beef cattle profitability.

## 3. Results and discussions

### 3.1. Descriptive results

**3.1.1. The socioeconomic characteristics of smallholder beef cattle farmers based on mean scores of continuous variables.** Table 3 results reveal that smallholder beef cattle farmers had an average age of 52.73 years. This suggests that farmers were of working age, which is vital in the successful implementation of intensive beef cattle production practices for more profit [18]. Farmers had an average family size of 7.07 members. This shows that farmers have a greater labor force opportunity for beef cattle production and sales operations, thereby increasing the profitability of the beef cattle farming business [18]. Moreover, the farmers had an average of 18.23 cattle herd size with 8.54 years of farming experience. It signifies that beef cattle production is pretty new in most farming households, necessitating the development of

**Table 3. Socioeconomic characteristics of smallholder beef cattle farmers based on mean scores of continuous variables.**

| Variables (N = 388) | Mean | Maximum | Minimum | Std.Deviation |
|---|---|---|---|---|
| Farmer's age | 52.73 | 70.00 | 33.00 | 9.37 |
| Family size | 7.07 | 12.00 | 3.00 | 2.07 |
| Cattle herd size (local breed) | 18.23 | 50.00 | 2.00 | 12.00 |
| Experience in farming | 8.54 | 19.00 | 2.00 | 4.64 |
| Land owned (ha) | 10.59 | 230.67 | 0.41 | 30.83 |

knowledge and expertise for increased efficiency. The findings of this study further indicated that the average land area possessed by beef cattle farmers was 10.59 hectares.

**3.1.2. The socioeconomic characteristics of smallholder beef cattle farmers based on frequency and percentage scores of categorical variables.** Table 4 results showed that men dominated beef cattle farming activities among smallholder beef cattle farmers. There were around 98.2% males and 1.8% females among the 388 sampled respondents. To eliminate

**Table 4. Socioeconomic characteristics of smallholder beef cattle farmers based on frequency and percentage scores of categorical variables.**

| Variables (N = 388) | Frequency | Percentage (%) |
|---|---|---|
| **Gender** | | |
| Male | 381 | 98.2 |
| Female | 7 | 1.8 |
| **Education level** | | |
| Primary Educ. | 27 | 7.0 |
| Secondary Educ. | 36 | 9.3 |
| College Educ. | 325 | 83.8 |
| **Access to farm credits** | | |
| Yes | 347 (277)* | 89.4 (71.4)* |
| No | 41 | 10.6 |
| **Access to market information** | | |
| Yes | 348 | 89.7 |
| No | 40 | 10.3 |
| **Access to veterinary services** | | |
| Yes | 347 | 89.4 |
| No | 41 | 10.6 |
| **Pasture availability and sufficient** | | |
| Yes | 321 | 82.7 |
| No | 67 | 17.3 |
| **Marketing channels (Market outlets)** | | |
| Open markets | 294 | 75.8 |
| Slaughterhouses/butcheries | 73 | 18.8 |
| Tajikistan Meat Commission | 7 | 1.8 |
| Middlemen | 14 | 3.6 |
| **Selling contract** | | |
| Yes | 340 | 87.6 |
| No | 48 | 12.4 |

* The figures in parentheses solely represent the frequency and percentage of farmers who utilized credits for cattle production

gender inequalities, females should indeed be motivated to participate in beef cattle production because practically all females are active in production activities, while males relocate from rural to urban areas in search of work [25, 29]. The findings also showed that the majority of farmers had a high degree of literacy. Around 83.8% of farmers had a college degree, while 7.0% and 9.3% had elementary and secondary education, respectively. Besides, around 89.4% of farmers had access to farm credits, whereas 10.6% did not. Despite this high level of access to farm credits, only 71.4% of farmers used farm credit points to produce beef cattle. Regarding the accessibility to veterinary services, the statistics reveal that 89.4% of farmers had access. This means that there is a high level of accessibility to veterinary care. This increase in access to veterinary services is critical for lowering the cost of obtaining consultations required for increased beef cattle production [25].

According to the findings, 89.7% of farmers had access to market information, while only 10.3 of farmers did not. This might imply that farmers were more accessible to accurate market knowledge, allowing 75.8% of farmers to sell their beef cattle to open market outlets (profitable outlets) rather than middlemen/traders/brokers. Farmers' revenues are eroded when beef cattle are sold to middlemen/traders/brokers. Accurate market information is critical for farmers looking to improve the performance (profitability) of their beef cattle business [18]. Additionally, about 82.7% of farmers acknowledged the availability of pasture for grazing, which encouraged 87.6% of farmers to get into a selling contract agreement. This helps to protect the interest of beef cattle. The abundance of grazing pasture is significant in beef cattle production, which increases the supply of beef cattle for sale [18].

## 3.2. Economic results

### 3.2.1. The profitability of beef cattle farming among smallholder beef cattle farmers.
Profitability is the primary aim of every agricultural business, therefore profitability is determined by calculating the gross margin (GM), and it measures the operating efficiency [30]. Table 5 shows that farmers had a mean GM of 353.77 US$ per beef cattle. This indicates that beef cattle production is a viable (profitable) enterprise in the study area. This is in line with the observations of Kibona and Yuejie [25], Mafimisebi et al. [31], Nasiru et al. [32], Okoruwa et al. [33], and Nkadimeng et al. [34], who found livestock farming to be a profitable venture.

**Table 5. The gross margin of beef cattle farming among smallholder beef cattle farmers (N = 388).**

| Parameters | The estimated value per cattle (US$) (N = 388) | | | |
|---|---|---|---|---|
| Cattle production variable costs | Mean | Max. | Min. | Std.Deviation |
| Husbandry labor* | 9.95(13.1)[1] | 119.90 | 0.00 | 21.04 |
| Medicines (treatment)* | 19.85(26.1) | 111.90 | 0.00 | 21.24 |
| Feeds (silage, fodder, soybean meal, & urea) * | 44.51(58.6) | 383.70 | 0.00 | 74.66 |
| Vitamins and mineral salts* | 0.51(0.7) | 6.40 | 0.00 | 1.51 |
| Sales and logistics | 1.11(1.5) | 6.20 | 0.00 | 1.90 |
| **Total variable cost (TVC)*** | **75.92** | **511.50** | **1.80** | **103.36** |
| **Earn revenue and sales value** | | | | |
| Selling price for cattle * | 429.69 | 680.00 | 50.00 | 138.58 |
| **Total revenue (TR) per cattle*** | **429.69** | **680.00** | **50.00** | **138.58** |
| **Gross margin(GM) = (Total revenue—Total cost)*** | **353.77** | **634.30** | **-356.60** | **231.75** |

*At the time of sale, beef cattle on average were 4.5 years old.

[1]The number in parentheses is the ratio of cost-share to total variable costs.

The findings showed that feed cost was the highest operational expenditure, accounting for around 58.6% of the total variable production costs. It was followed by medications (treatment expenses) (26.1%), and husbandry labor costs (13.1%). Other expenditures were just under 2%, including vitamin and minerals salt costs (0.7%) and sales and logistics costs (1.3%). This suggests that the Livestock and Pasture Development Project I & II (LPDP I & II) undertaken in the research region had little impact on feed cost reduction; however, the potential to increase beef cattle farming profitability is great once feed costs are minimized. This implies that a decrease in production costs as a result of pasture availability (free or cheap available) would result in high profitability [25]. In addition, the significant cost of treatment or pharmaceuticals is a result of the incidence of animal illnesses in the research area. According to kibona and Yuejie [25], treating beef cattle following an outbreak of the illness is expensive. As a result, developing beef cattle-specific healthcare initiatives to address possible health issues is the greatest and only method to minimize cost; consequently increasing profitability [25].

**3.2.2. Factors that influence profitability of beef cattle farming among smallholder beef cattle farmers.** Table 6 presents the results of a multiple regression analysis utilizing OLS on the key parameters (variables) that determined the profitability of beef cattle farming among smallholder beef cattle farmers. The model was calculated utilizing SPSS v.22, and it fitted well and was statistically significant at *P < 0.05*. The model's adjusted R—squared = 0.833, signifying that the independent variables explain 83.3% of the variance in profitability per beef cattle among beef cattle farmers.

Holding other factors constant, variable costs (feed cost and sales and logistics costs) had a negative and statistically significant impact on profitability. A negative correlation reveals that as feed and logistics costs increase by 1US$, profitability per beef cattle falls by 0.713% and 0.148%, respectively. Moreover, the negative values for sales and logistics indicated that a rise in these parameters would result in a fall in interviewees' profits, which might be attributable

**Table 6. Multiple linear regression estimates using ordinary least squares (OLS) on the factors influencing the profitability of beef cattle farming among smallholder beef cattle farmers N = 388).**

| Variables | Coefficients (β) | Std. Error |
|---|---|---|
| Feed costs | -0.713* | 0.000 |
| Husbandry labor costs | 0.205 | 0.001 |
| Medicines(treatment) costs | -0.072* | 0.000 |
| Sales and logistics costs | -0.148* | 0.018 |
| Education level | 0.066* | 0.010 |
| Family size | 0.072* | 0.004 |
| Land owned | 0.084* | 0.054 |
| Cattle herd size | -0.107* | 0.001 |
| Experience in farming | 0.196* | 0.002 |
| Access to veterinary services | 0.142 | 0.062 |
| Access to credits | 1.369* | 0.179 |
| Pasture availability | 0.215* | 0.016 |
| Selling contract | -1.015* | 0.197 |
| Marketing channels (market outlets) | 0.002 | 0.010 |
| Access to market information | 0.013 | 0.053 |
| **Constant** | **91.729*** | **0.104** |
| **R Squared (R²)** | **0.840(84.0%)** | |
| **Adjusted R squared (Adj.R²)** | **0.833(83.3%)** | |

*Indicate statistical significance level at 5% (P < 0.05).

to the expensive cost of transportation in the research location. This observation is consistent with those of Nasiru [35] and Suleiman et al [36]. Furthermore, feed expenses, as well as sales and logistics costs, highlight the capital involved in the production and trading operations. Marketing expenditures, on the other hand, indicate apparent transaction fees that may impede profit maximization [37].

The coefficients for medications (treatment costs) had a negative influence on the profitability of beef cattle farming and were statistically significant at $P < 0.05$. A positive relation for medication cost indicates that cash invested in pharmaceuticals affects beef cattle profitability. This means that for every \$1 increase in medication costs, profitability falls by 0.072%. It is expensive to treat beef cattle after a disease outbreak [25]. Thus, implementing beef cattle-specific health programs to address potential health issues is the best and only way to maximize profit [25].

The educational coefficient is positively significant. This validates an evident finding that almost educated farmers are quite cost-effective, and is also influential on the socio-cultural capital impacts that literacy may assist to organize. Education improves the ability to apply knowledge and the usage of ideas essential for cattle production [38]. The variable is also a predictor of the acceptance of advancements and new technologies required to boost farm output [37]. This finding contradicts the findings of Nkadimeng et al. [34], who found a negative relationship between a farmer's education and profit earned. The study did explain, however, that more educated farmers may have other income-generating sources other than the beef cattle farming project, attempting to put less emphasis on beef cattle profit maximization [34].

The profitability of beef cattle production was positively and significantly related to family size. Huge families have the opportunities for much more family labor involved in production and sales operations, thereby boosting profitability. The number of people responsible for the day-to-day care of cattle is measured by labor. Farmers that have the personnel to care for the cattle can manage higher herd sizes, which increases the chances of improving profitability [37].

Given that grazing area, abundance is significant in beef cattle productivity and profitability, which improves the economic opportunities and effectiveness of beef cattle production [18]. This study also found that land ownership had a positive impact on the profitability of beef cattle production, and it was statistically significant at the 5% significance level. This implies that when land ownership rises in rural areas, beef cattle profitability rises. This result is consistent with the findings of Nkadimeng et al. [34].

Beef cattle herd size had a negative influence on beef cattle production profit yet was statistically significant at a 5% level of significance. The findings show that as herd size grows the profitability of beef cattle declines; it is difficult and expensive to manage larger herd sizes effectively. This finding is in line with koknaroglu et al., [39], who reported that fewer beef cattle per farm result in higher profitability per animal. Therefore, farmers should be motivated to minimize massive cattle herds, stay in grazing areas, and efficiently feed beef cattle before selling (production efficiency) for a higher return [18]. A profit-maximizing firm in agricultural production can produce and distribute superior products in the target marketplace at a financial benefit for the enterprise's existence [40].

The coefficient for farming experience was significantly positive at the 5% level. This indicates that farmers who have been raising beef cattle for a long time generated higher money as a result of their understanding of trends in beef cattle production and marketing. This result confirms Afolabi's previous results [41]. Farming experience highlights the power of social media platforms and linkages reinforced to improve the exploration of new clients [42]. Furthermore, farming experience provides information and producing practices that are recognized to be effective in production and marketing operations [43].

Exposure to farm credits had a positive impact on the profitability of beef cattle production, and it was statistically significant at a 5% level. The positive indication suggests that as a farmer's exposure to farm credits increases, so does the profitability of beef cattle production. The availability of finances for agribusiness (commercial production) is captured by access to farm credit [37]. Furthermore, credit facilitates the purchasing of inputs, extension trips, and other assistance programs for diversified business activities, hence enhancing profitability [37].

Furthermore, the abundance of grazing resources for beef cattle production is represented by pasture accessibility [37]. Because stocking rates are not limited in rural areas, farmers' perceptions of abundant grazing pastures motivate them to participate in production operations, enhancing farm operating efficiency [37]. This study also revealed that pasture availability had a positive and significant impact on beef cattle profitability. This indicates that when the grazing area expands, the profit per cattle significantly increases.

Additionally, selling contracts provide farmers with a secured market at the local retail outlet, minimizing marketing and distribution expenditures as well as price risk. However, the results of the present study revealed that participating in selling contracts was significantly but negatively correlated with the profitability of beef cattle production and was statistically significant at the 5% level of significance. This implies that a farmer, as a weaker partner, is vulnerable to exploitation by the buying company [44]. If farmers have committed significant investment into resources related to the contract commodities which are perishable and not conducive to translation into perishable items on the farm, a company can take monopoly rent in the output market. In addition, other significant negative externalities which hinder profitability are the danger of farmers engaging in extra-contractual sales, particularly when the agreed price is set and the stock value at the time of the sale is greater than the agreed price [44]. In general, selling contracts should be mutually beneficial (flexible and profitable) to increase profitability.

## 4. Conclusion

This study assessed the economic benefits of beef cattle farming and its determinants at the smallholder farmer's level. As per the descriptive analyses, the majority of farmers had a high level of literacy with access to farm credits, market information, pasture, and veterinary services, resulting in high profitability (gross margin) of beef cattle farming. Meanwhile, econometric estimation showed that the profitability of beef cattle farming among farmers was significantly influenced by education level, family size, farming experience, pasture availability, land size owned, selling contract, feed costs, medications expenses, access to credits, and sales costs. This signifies that beef cattle production is a feasible (profitable) business in the study area. However, the potential for increased profitability is significant if existing resources are efficiently coordinated and production expenses, notably feed and healthcare costs, are minimized. The results have diverse implications in terms of what components need to be handled to increase the profitability of beef cattle production among Tajik smallholder farmers. Thus, the government should develop additional measures for addressing concerns such as capacity building, suitable and freely available pasture as well as health management, to boost beef cattle profitability among smallholder beef cattle farmers in Tajikistan. Due to restricted funds, this research was restricted to only one district rather than all districts covered by the project. The extent of market participation among smallholder farmers should be researched further to identify their level of beef cattle commercialization.

## Supporting information

**S1 File.**
(PDF)

## Author Contributions

**Conceptualization:** Zhang Yuejie.

**Data curation:** Farrukh Jobirov.

**Formal analysis:** Farrukh Jobirov.

**Funding acquisition:** Zhang Yuejie.

**Investigation:** Farrukh Jobirov.

**Methodology:** Cornel Anyisile Kibona.

**Project administration:** Zhang Yuejie.

**Resources:** Farrukh Jobirov.

**Software:** Zhang Yuejie.

**Supervision:** Zhang Yuejie.

**Validation:** Cornel Anyisile Kibona.

**Visualization:** Zhang Yuejie.

**Writing – original draft:** Farrukh Jobirov.

**Writing – review & editing:** Cornel Anyisile Kibona.

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
