## [Decision Letter · Decision Letter 0]

15 Jun 2022

PONE-D-21-37158

Evaluating Profitability of Beef Cattle Farming and its Determinants among Smallholder Beef Cattle Farmers in the Baljovan District of Khatlon Region, Tajikistan

PLOS ONE

Dear Dr. Jobirov,

Thank you for submitting your manuscript to PLOS ONE. After careful consideration, we feel that it has merit but does not fully meet PLOS ONE’s publication criteria as it currently stands. Therefore, we invite you to submit a revised version of the manuscript that addresses the points raised during the review process.

Dear Sir, there are the reviewers. Please adjust your paper and send it back.

Best regards

We look forward to receiving your revised manuscript.

Kind regards,

Julio Cesar de Souza, Ph.D.

Academic Editor

PLOS ONE

**Journal requirements:**

2. During our internal checks, the in-house editorial staff noted that you conducted research or obtained samples in another country. Please check the relevant national regulations and laws applying to foreign researchers and state whether you obtained the required permits and approvals. Please address this in your ethics statement in both the manuscript and submission information. In addition, please ensure that you have suitably acknowledged the contributions of any local collaborators involved in this work in your authorship list and/or Acknowledgements. Authorship criteria is based on the International Committee of Medical Journal Editors (ICMJE) Uniform Requirements for Manuscripts Submitted to Biomedical Journals - for further information please see here: https://journals.plos.org/plosone/s/authorship.

“This research is supported and funded by the National Beef Cattle Industrial Technology System and Industrial Economy Research Project under the Ministry of Agriculture in the People’s Republic of China (PRC) (CARS-37).”

“Yes;

This research is supported and funded by the National Beef Cattle Industrial Technology System and Industrial Economy Research Project under the Ministry of Agriculture in the People’s Republic of China (PRC) (CARS-37).

**Additional Editor Comments:**

Dear Sir, there are review suggestions.

Please adjust and send it back.

Best regards.

Julio Souza

Reviewers' comments:

Reviewer's Responses to Questions

**Comments to the Author**

1. Is the manuscript technically sound, and do the data support the conclusions?

Reviewer #1: Partly

Reviewer #2: Yes

Reviewer #3: Yes

2. Has the statistical analysis been performed appropriately and rigorously? 

Reviewer #1: No

Reviewer #2: Yes

Reviewer #3: Yes

3. Have the authors made all data underlying the findings in their manuscript fully available?

Reviewer #1: Yes

Reviewer #2: Yes

Reviewer #3: Yes

4. Is the manuscript presented in an intelligible fashion and written in standard English?

Reviewer #1: Yes

Reviewer #2: No

Reviewer #3: Yes

5. Review Comments to the Author

Reviewer #1: Table 4: Results in this Table seem to be incorrect: I’m not convinced that cattle herd size, land owned and pasture availability have a negative influence on beef cattle production profit. In addition, I’m not sure that medicines costs has a positive influence on beef cattle production profit.

Reviewer #2: Farrukh et al evaluated the profitability of beef cattle farming and its determinants among smallholder farmers in the Baljovan district of Khatlon Region, Tajikistan. The issues addressed in the study seem interesting in solving the challenges of poverty at the smallholder level in Tajikistan. However, the manuscript was written poorly. Detailed comments are given inside the manuscript file uploaded herewith.

Reviewer #3: The primary goal of this paper is to evaluate the profitability of beef cattle farming and its determinants among smallholder beef cattle farmers in the Baljovan District of Khatlon Region, Tajikistan.

Because the topic is somewhat unique, I accept it as an original submission for publication. While I believe the manuscript provides a solid foundation for scientific communication, I believe the following changes should be made before publication:

General comments

-There are minor grammar errors and changes suggested in the attached.

Materials and Methods section;

-worth to consider starting the section with the theoretical framework before section 2.5 of data analysis, it should be arranged as; 2.4. Theoretical framework, followed by 2.5.Data analysis.

-Indicate the variable units of measurement, explanations/justifications, and presumed signs.

Conclusions section

- Implications and limitations of the research should be indicated, as well as the areas of further study.

6. PLOS authors have the option to publish the peer review history of their article (what does this mean?). If published, this will include your full peer review and any attached files.

Reviewer #1: No

Reviewer #2: No

Reviewer #3: No

---

## [Author Response · Author response to Decision Letter 0]

23 Jul 2022

Response to the Reviewers’ Comments 

We are grateful to the editor and reviewers for reading our manuscript and providing detailed and constructive feedback. Below is a point-by-point response to your comments and recommendations, as well as how each one was addressed in the revision.

Response to the Editors’ Comments

Comment 1: Please ensure that your manuscript meets PLOS ONE's style requirements, including those for file naming.

Response: We appreciate the editor's reminder. We double-checked and fulfilled all rules to ensure that our manuscript meets PLOS ONE standards.

Comment 2: During our internal checks, the in-house editorial staff noted that you conducted research or obtained samples in another country. Please check the relevant national regulations and laws applying to foreign researchers and state whether you obtained the required permits and approvals. Please address this in your ethics statement in both the manuscript and submission information. In addition, please ensure that you have suitably acknowledged the contributions of any local collaborators involved in this work in your authorship list and/or Acknowledgements. Authorship criteria is based on the International Committee of Medical Journal Editors (ICMJE) Uniform Requirements for Manuscripts Submitted to Biomedical Journals - for further information please see here: https://journals.plos.org/plosone/s/authorship.

Response: We appreciate the editor's comments. Firstly, I am from Tajikistan; therefore, Tajikistan is not a foreign country for me. Secondly, as a Ph.D. student at Jilin Agriculture University-China, my study field is Tajikistan, specifically the Baljovan District in the Khatlon Region. We observed all applicable national rules and got all necessary permits throughout data collection, as stated in ethics declarations. Furthermore, throughout data gathering, all local contributors were acknowledged.

Comment 3: Thank you for stating the following in the Funding Section of your manuscript:

“This research is supported and funded by the National Beef Cattle Industrial Technology System and Industrial Economy Research Project under the Ministry of Agriculture in the People’s Republic of China (PRC) (CARS-37).”

“Yes;

This research is supported and funded by the National Beef Cattle Industrial Technology System and Industrial Economy Research Project under the Ministry of Agriculture in the People’s Republic of China (PRC) (CARS-37).

Response: We appreciate the editor's comments. We apologize for failing to provide financial details in our funding statement. In response, we have provided our revised statements in our cover letter.

Additional Editor Comments: Dear Sir, there are review suggestions. Please adjust and send it back.

Response: We appreciate the editor's updates. We have read and reacted to all of the editor's and reviewers' remarks and suggestions. We are currently submitting it to you for further action.

Response to Reviewer 1 Comments

Comment 1: Table 4: Results in this Table seem to be incorrect: I’m not convinced that cattle herd size, land owned, and pasture availability have a negative influence on beef cattle production profit. In addition, I’m not sure that medicines costs have a positive influence on beef cattle production profit.

Response: We thank the reviewer for these comments. We concur with the reviewer that some of the results in Table 4 (now Table 6) appear to be inaccurate. We apologize for the mistakes. We double-checked and discovered that both land ownership and pasture availability had a positive impact on beef cattle probability, but medicine costs had a negative influence. However, beef cattle herd size continued to have a negative influence, as mentioned in the text, since the huge stock of beef cattle may be difficult to maintain for smallholders, resulting in low profitability per cattle.

Response to Reviewer 2 Comments

Comment 1: Farrukh et al evaluated the profitability of beef cattle farming and its determinants among smallholder farmers in the Baljovan district of Khatlon Region, Tajikistan. The issues addressed in the study seem interesting in solving the challenges of poverty at the smallholder level in Tajikistan. However, the manuscript was written poorly. Detailed comments are given inside the manuscript file uploaded herewith.

Response: We thank the reviewer for this comment. We apologize for disappointing our esteemed reviewer with the poor writing in our prior manuscript; all recommendations have been addressed in our amended version. We also appreciate the reviewer's comment that the issues addressed in the study appear to be interesting in addressing the concerns of poverty at the smallholder level in Tajikistan. We trust the updated text will satisfy the reviewer.

Responses to Specific Comments in the annotated PDF from Reviewer 1

Comment 3: for the smallholder farmers to alleviating poverty.

Response: We appreciate this comment from the reviewer. We agree with the reviewer that the phrase should read as “for the smallholder farmers to alleviating poverty”.

Comment 4: try to rewrite this phrase in a more meaningful way.

Response: We thank the reviewer for this comment. We have rephrased the sentence in a more meaningful way. Currently, the phrase read as “the smallholder beef cattle farmers are known for making little profits” instead of smallholder beef cattle farmers are notorious for inefficient profit maximization.

Comment 5: data collected using questionnaires.

Response: We appreciate this comment from the reviewer. We agree with the reviewer that the phrase should read as “data collected using questionnaires”.

Comment 6: why do you repeat the same word?.

Response: We appreciate the reviewer's comments. To minimize repetition, we checked and removed the word farmers. 

Comment 7: Delete the word “that”.

Response: We thank the reviewer for this comment. We have deleted the word. 

Comment 8: Stop the sentence here and start the next sentence by using connector words.

Response: We thank the reviewer for this comment. We appreciate the help provided by the reviewer in correcting the mistakes in our manuscript. We have added stop and started the next sentence by using connector words.

Comment 9: If you think the three words are the same, choose one of them to make the sentence more formal. 

Response: We appreciate the comment from the reviewer. We have followed the suggestions from the reviewer to choose one to make the sentence more formal.

Comment 10: Meanwhile, the profitability of beef farming among farmers was positively and significantly influenced by education level, family size, farming experience, medications expenses, and access to credits (P<0.05).

Response: We thank the reviewer for this comment. We appreciate the help provided by the reviewer in correcting the abstract section. We have followed the suggestions from the reviewer to present the results “Meanwhile, the profitability of beef farming among farmers was positively and significantly influenced by education level, family size, farming experience, medications expenses, and access to credits (P<0.05)”.

Comment 11: Reference No. 2 has been excessively cited. 

Response: We thank the reviewer for this comment. Reference No. 2 is a government document that contains extensive information regarding Tajik Livestock and Pasture Development Projects I and II (LPDP I & II). As a result, we had to quote the original document for any project-related issues. 

Comment 12: What influence? Positive or negative? describe it, please. 

Response: We appreciate the comment from the reviewer. We have described it as having a negative impact on the beef cattle industry.

Comment 13: the ideas before and after this word do not support this word. 

Response: We appreciate the reviewer's comments. We apologize for any ambiguous sentences. To produce a meaningful statement, we used the word "however" instead of "furthermore".

Comment 14: Investment initiatives to address the challenges of improving beef... 

Response: We appreciate this comment from the reviewer. We agree with the reviewer that the phrase should read as “Investment initiatives to address the challenges of improving beef cattle efficiencies” instead of “Investing in initiatives to address the challenges to improved beef cattle efficiencies”.

Comment 15: a relatively!

Response We appreciate this comment from the reviewer. We agree with the reviewer that the phrase should read as “a relatively” instead of “relative”.

Comment 16: make it two sentences, it's too long!

Response: We thank the reviewer for this comment. We have rephrased the sentence based on the comments from the reviewer.

Comment 17: .was to alleviate...

Response: We thank the reviewer for this comment. We agree with the reviewer that the phrase should read as “The project's overall purpose was to alleviate poverty in Khatlon Region.

Comment 18: not needed!

Response: We thank the reviewer's comments. As indicated by the reviewer, we eliminated the term "furthermore”.

Comment 19: write the name of the project here, because you are in a new paragraph.

Response: We thank the reviewer for this observation. We wrote the name of the project as suggested by the reviewer in the enclosed PDF file.

Comment 20: are you trying to say for three purposes? make it clear

Response: We thank the reviewer for this comment. We have made it clear by specifying three purposes.

Comment 21: better to put it after press release

Response: We thank the reviewer for the observation. We agree with the reviewer. We have put the citation after press release as suggested by the reviewer in the enclosed PDF file. 

Comment 22: stop the sentence here and start the next like this: The first was to realize institutional advancement by strengthening the public sector and civic groups, such as through developing reliable and successful prop-poor grassland monitoring systems. then write as second, third... 

Response: We thank the reviewer for this comment. We have rephrased the sentences as suggested by the reviewer in the enclosed PDF file. Thus, the rephrases read as “The first was to realize institutional advancement by strengthening the public sector and civic groups, such as through developing reliable and successful prop-poor grassland monitoring systems. The second was to improve animal health and performance by improving access to veterinary care, which resulted in lower morbidity and improved herd performance. The third purpose was to achieve grassland advancement and expansion for disaster mitigation by improving access to more capable and climate-resilient grassland zones, as well as diverse revenue possibilities for beef cattle societies, through self-sustaining, community-led biodiversity conservation. The major aim was to improve the living standards of beef cattle farmers in selected districts of the Khatlon region and the nation as a whole.

Comment 22: The major aim was to improve the living standards of beef cattle farmers..... 

Response: We have rephrased the sentence based on the comments from the reviewer. The phrase now reads as “The major aim was to improve the living standards of beef cattle farmers in selected districts of the Khatlon region and the nation as a whole”.

Comment 23: misused

Response: We thank the reviewer for this comment. We are sorry for misusing the word “nevertheless”. We have replaced it with the word “still”.

Comment 24: stage

Response: We thank the reviewer for this comment. The word has been rephrased as suggested by the reviewer.

Comment 25: not needed

Response: Thank you for the observation. We have deleted the word enterprise as suggested by the reviewer in the enclosed PDF file.

Comment 26: delete,

Response: We thank the reviewer for this comment. We have deleted the (,) as suggested by the reviewer in the enclosed PDF file.

Comment 27: for this study.

Response: We thank the reviewer for this comment. We have rephrased the phrase to read “for this study” as suggested by the reviewer in the enclosed PDF file. 

Comment 28: human or cattle 

Response: We thank the reviewer for this comment. We have added the word “people” to indicate human and not cattle.

Comment 29: good if it is mentioned first (2.1).

Response: We thank the reviewer for the comment. The highlighted section has been moved to section 2.1 as suggested by the reviewer in the enclosed PDF file.

Comment 30: not needed in the title/sub-title...

Response: We thank the reviewer for the comment. The highlighted word has been removed in the title/sub-title as suggested by the reviewer in the enclosed PDF file.

Comment 31: You already abbreviated it, use the abbreviated form throughout except at the beginning of the sentence!

Response: We appreciate the reviewer's insightful observation. We have deleted the word accordingly.

Comment 32 abbreviated already!

Response: We thank the reviewer for this comment. We have deleted the word accordingly.

Comment 33: abbreviated already

Response: We thank the reviewer for this comment. The word has been removed as suggested by the reviewer in the enclosed PDF file.

Comment 34: not needed

Response: We thank the reviewer for the comment. We agree with the reviewer. The highlighted sentence in the enclosed PDF file has been deleted.

Comment 35: not needed.

Response: We thank the reviewer for the comment. We agree with the reviewer. The highlighted sentence in the enclosed PDF file has been deleted.

Comment 36: not needed.

Response: We thank the reviewer for the comment. We agree with the reviewer. The highlighted sentence in the enclosed PDF file has been deleted.

Comment 37 not needed.

Response We appreciate the reviewer's comments. We concur with the reviewer. In the enclosed PDF file, the highlighted word (investigation) has been removed.

Comment 38: Highlighted word (found)

Response: We thank the reviewer for the comment. We have rewritten the word “found” to mean “showed”.

Comment 39: Besides, around...

Response: We thank the reviewer for the comment. The sentence has been rephrased as suggested by the reviewer in the enclosed PDF file.

Comment 40: Make it two tables. One continues variables and two, categorical variables. add N in column for each variable.

Response: We thank the reviewer's comments. We created two tables: one for continuous variables and one for categorical variables. Furthermore, as advised by the reviewer, we have inserted N in the column for each variable.

Comment 41: The highlighted word (investigation) 

Response: We appreciate the reviewer's comments. We concur with the reviewer. In the enclosed PDF file, the highlighted word (investigation) has been removed.

Comment 42: why don't you use one word? there are several repeated words and phrases in the text, under ().

Response: We thank the reviewer for this comment. The highlighted section has been rewritten as suggested by the reviewer in the enclosed PDF file. We have chosen one word (profitability) to be used.

Comment 43: The findings showed that feed cost was the highest operational expenditure...

Response: We thank the reviewer for the comment. The highlighted words have been rewritten as suggested by the reviewer to read “The findings showed that feed cost was the highest operational expenditure” to make it clear.

Comment 44: use OLS only

Response: We thank the reviewer for the comment. We concur with the reviewer. We have used OLS only.

Comment 45: which variables? make it clear! use comma appropriately

Response: We thank the reviewer. We agree with the reviewer. We have rephrased the sentence to read “A negative correlation reveals that as feed and logistics costs increase by 1US$, profitability per beef cattle falls by 0.713% and 0.148%, respectively.

Comment 46: This is too exaggerated. You are talking about smallholder farmers (not commercial farmers), do they have access for commercial feeds or is it profitable or even affordable?.

Response: We thank the reviewer for the comment. We are very sorry for this confusion. We double-checked and rephrased the sentence to read This study also revealed that pasture availability had a positive and significant impact on beef cattle profitability. This indicates that when the grazing area expands, the profit per cattle significantly increases.

Comment 47: not clear

Response: We thank the reviewer for this comment. We are sorry for the disappointment. We have rephrased the sentence to make it clear. 

Comment 48: add a reference that support your claim of poor nutritional values of pasture grasses in your study areas.

Response: We appreciate the reviewer for this comment. We are sorry for the disappointment. We rephrased the sentence to make it more understandable.

Comment 49: correct the grammar

Response: We thank the reviewer for this close observation. We are sorry for the poor grammar. We have collected the grammar. Currently, the sentence read as “additionally, selling contracts provide farmers with a secured market at the local retail outlet, minimizing marketing and distribution expenditures as well as price risk”.

Comment 50: present study ...

Response: We thank the reviewer for the comment. We have agreed, that the sentence has been rephrased using the phrase “present study” as suggested by the reviewer in the enclosed PDF file.

Comment 51: but negatively

Response: We thank the reviewer for the comment. The sentence has been rephrased using the phrase “but negatively” as suggested by the reviewer in the enclosed PDF file

Comment 52: which company you are referring to?.

Response: We thank the reviewer for this comment. We are sorry for this confusion. The company we refer to here is a buying company (contracted with farmers to buy beef cattle).

Comment 53: "In general" is better.

Response: We thank the reviewer for this important comment. We have followed the suggestions from the reviewer to use the word “in general” instead of “as a results”.

Comment 54: this sentence was mentioned three times( abstract, introduction, and conclusion). Try to write the idea in different ways!. Why you repeat all these? It would have been good if you had started the conclusion as follow: This study assessed the economic benefits of beef cattle farming and its determinants at the smallholder farmers level. then continue to major findings.

Response: We thank the reviewer for this comment. We are very sorry for the repetitions. We have rewritten the whole conclusion to make it clear. Currently, the conclusion read “This study assessed the economic benefits of beef cattle farming and its determinants at the smallholder farmer's level. As per the descriptive analyses, the majority of farmers had a high level of literacy with access to farm credits, market information, pasture, and veterinary services, resulting in high profitability (gross margin) of beef cattle farming. Meanwhile, econometric estimation showed that the profitability of beef cattle farming among farmers was positively and significantly influenced by education level, family size, farming experience, pasture availability, land size owned, selling contract, feed costs, medications expenses, access to credits, and sales costs. This signifies that beef cattle production is a feasible (profitable) business in the study area. However, the potential for increased profitability is significant if existing resources are efficiently coordinated and production expenses, notably feed and healthcare costs, are minimized. The results have diverse implications in terms of what components need to be handled to increase the profitability of beef cattle production among Tajik smallholder farmers. Thus, the government should develop additional measures for addressing concerns such as capacity building, suitable and freely available pasture as well as health management, to boost beef cattle profitability among smallholder beef cattle farmers in Tajikistan. Due to restricted funds, this research was restricted to only one district rather than all districts covered by the project. The extent of market participation among smallholder farmers should be researched further to identify their level of beef cattle commercialization.

Response to Reviewer 2 Comments

General Comments

Comment 1: The primary goal of this paper is to evaluate the profitability of beef cattle farming and its determinants among smallholder beef cattle farmers in the Baljovan District of Khatlon Region, Tajikistan.

Because the topic is somewhat unique, I accept it as an original submission for publication. While I believe the manuscript provides a solid foundation for scientific communication, I believe the following changes should be made before publication.

Response: We thank the reviewer for seeing the useful information contained in our manuscript and consider it an interesting article. We have improved our manuscript based on the reviewers’ comments. We hope the reviewer will find it more useful. 

Comment 2: -There are minor grammar errors and changes suggested in the attached.

Response: We thank the reviewer for this comment. We are sorry for the shortcomings in our previous manuscript. We diligently and carefully corrected grammar errors and changes suggested in the attachment to enable simple reading.

Materials and Methods section;

Comment 3: -worth considering starting the section with the theoretical framework before section 2.5 of data analysis, it should be arranged as; 2.4. Theoretical framework, followed by 2.5.Data analysis.

Response: We thank the reviewer for this suggestion. The theoretical framework has been moved to section 2.5 as suggested by the reviewer.

Comment 4: -Indicate the variable units of measurement, explanations/justifications, and presumed signs.

Response: We thank the reviewer for this comment. We have created a table of variables including units of measurement, explanations, and presumed signs. We hope it is clear now.

Conclusions section

Comment 5 Implications and limitations of the research should be indicated, as well as the areas of further study.

Response: We thank the reviewer for this comment. We have indicated the implications and limitations of the research, as well as the areas of further study. The implication, the results have diverse implications in terms of what components need to be handled to increase the profitability of beef cattle production among Tajik smallholder farmers. Limitations of the research, Due to restricted funds, this research was restricted to only one district rather than all districts covered by the project. The areas of further study, the extent of market participation among smallholder farmers should be researched further to identify their level of beef cattle commercialization.

---

## [Editor Report · Decision Letter 1]

28 Aug 2022

Evaluating Profitability of Beef Cattle Farming and its Determinants among Smallholder Beef Cattle Farmers in the Baljovan District of Khatlon Region, Tajikistan

PONE-D-21-37158R1

Dear Dr. Jobirov,

We’re pleased to inform you that your manuscript has been judged scientifically suitable for publication and will be formally accepted for publication once it meets all outstanding technical requirements.

Kind regards,

Julio Cesar de Souza, Ph.D.

Academic Editor

PLOS ONE
---

## [Editor Report · Acceptance letter]

5 Sep 2022

PONE-D-21-37158R1 

Evaluating Profitability of Beef Cattle Farming and its Determinants among Smallholder Beef Cattle Farmers in the Baljovan District of Khatlon Region, Tajikistan 

Dear Dr. Jobirov:

I'm pleased to inform you that your manuscript has been deemed suitable for publication in PLOS ONE. Congratulations! Your manuscript is now with our production department. 

Kind regards, 

on behalf of

Dr. Julio Cesar de Souza 

Academic Editor

PLOS ONE